# Cell formation and layout design using genetic algorithm and TOPSIS: A case study of Hydraulic Industries State Company

Dhulfiqar Hakeem Dhayef[1,2]*, Sawsan S. A. Al-Zubaidi[2], Luma A. H. Al-Kindi[2], Erfan Babaee Tirkolaee[3,4,5]

1 Department of Cooling and Air Conditioning Engineering, Imam Ja'afar Al-Sadiq University, Baghdad, Iraq, 2 Department of Production Engineering & Metallurgy, University of Technology, Baghdad, Iraq, 3 Department of Industrial Engineering, Istinye University, Istanbul, Turkey, 4 Department of Industrial Engineering and Management, Yuan Ze University, Taoyuan, Taiwan, 5 Department of Industrial and Mechanical Engineering, Lebanese American University, Byblos, Lebanon

* pme.20.37@grad.uotechnology.edu.iq

## Abstract

Cell formation (CF) and machine cell layout are two critical issues in the design of a cellular manufacturing system (CMS). The complexity of the problem has an exponential impact on the time required to compute a solution, making it an NP-hard (complex and non-deterministic polynomial-time hard) problem. Therefore, it has been widely solved using effective meta-heuristics. The paper introduces a novel meta-heuristic strategy that utilizes the Genetic Algorithm (GA) and the Technique of Order Preference Similarity to the Ideal Solution (TOPSIS) to identify the most favorable solution for both flexible CF and machine layout within each cell. GA is employed to identify machine cells and part families based on Grouping Efficiency (GE) as a fitness function. In contrast to previous research, which considered grouping efficiency with a weight factor ($q = 0.5$), this study utilizes various weight factor values (0.1, 0.3, 0.7, 0.5, and 0.9). The proposed solution suggests using the TOPSIS technique to determine the most suitable value for the weighting factor. This factor is critical in enabling CMS to design the necessary flexibility to control the cell size. The proposed approach aims to arrange machines to enhance GE, System Utilization (SU), and System Flexibility (SF) while minimizing the cost of material handling between machines as well as inter- and intracellular movements (TC). The results of the proposed approach presented here show either better or comparable performance to the benchmark instances collected from existing literature.

## Introduction

Businesses in today's highly competitive marketplaces understand the importance of responding rapidly to customer requests. This can only be accomplished if businesses are committed to a culture of constant improvement and quickly adapt to shifting customer demands. Cellular Manufacturing Systems (CMSs) are one such approach that is becoming the most

**Data Availability Statement:** Data relevant to this study are available from Zenodo at DOI:10.5281/zenodo.10147377 (https://zenodo.org/records/10147377).

**Funding:** The author(s) received no specific funding for this work.

**Competing interests:** The authors have declared that no competing interests exist.

important manufacturing mode because they allow for a great deal of adaptability without sacrificing efficiency. Manufacturing with CMS is efficient because it combines the adaptability of job shops with the efficiency of flow lines. As a concept of Group Technology (GT), CMS has proven to be one of the most effective ways to enhance productivity and flexibility [1, 2]. According to the literature, CMSs have various advantages, including reducing work-in-process inventories, throughput times, setup times, and response times to customer orders [3]. Cell Formation (CF) is the primary concern while designing the CMS. CF refers to the process of identifying part families and machine cells during the design stage of CMS. Essentially, manufacturing cells are groups of different machines that are physically close together and are capable of processing an entire family of parts [4, 5]. Once the part family and machine cell are determined, they are allocated into the available spaces using a machine cell layout. This strategy is particularly useful in batch-type production industries as it enhances productivity and flexibility [6]. Increased agility and competitiveness can be achieved through manufacturing systems with greater operational flexibility. In this regard, the term operational flexibility encompasses two key dimensions: demand flexibility and routing flexibility. Demand flexibility refers to the capacity of a cellular system to adjust to shifts in part mix and part demand. In other words, it facilitates the system to promptly and effectively adapt to alterations in customer requirements and production schedules. On the other hand, routing flexibility relates to the ability of a cellular system to process parts on different machine cells, thereby providing the necessary versatility to handle diverse product lines and production requirements. Companies can better respond to changing market conditions, maximize resource usage, and boost profits by building demand and route flexibility into their production processes [7].

The operational benefits of the CMSs are often praised, but the system's perceived lack of operational flexibility has been criticized [8, 9]. As an illustration of this criticized feature, CMS produces inefficient solutions and unstable machine utilization as a result of dynamic and random variations in part demands, raising worries about the management implications of flexible manufacturing. The design of a fixed cell layout can have a major impact on performance, especially in terms of machine utilization, but it also reduces flexibility [10]. As highly flexible manufacturing cells become increasingly commonplace, production managers are faced with the challenge of determining the best layout design for CMS. One of the most significant considerations in layout design is layout flexibility. It is crucial for the cell's enhanced adaptability to size changes and novel products [11].

Despite the paramount significance of CMS adoption, the requirement for layout design flexibility is not well addressed by current optimization models. However, efficient CMS design also requires forethought into CF design, in addition to the flexible layout design. This paper proposes a novel hybrid approach for interactive CF that offers high flexibility in cell layout. The suggested approach attempts to reduce the cost of material handling between machines, and inter- and intra-cellular movements while maximizing group efficiency, system flexibility, and utilization. Genetic Algorithm (GA) and Technique of Order Preference Similarity to the Ideal Solution (TOPSIS) are utilized to find the optimal design for CF and the layout of CMS. This hybrid approach maximizes group efficiency, system flexibility, and utilization while reducing handling costs. To the best of our knowledge, this is the first time a flexible layout design has been used to develop a close-to-ideal solution design by altering the weight factor in conjunction with a powerful evolutionary algorithm enhanced by a matrix-based chromosome structure. The TOPSIS method is then used to determine the ideal architecture for machine cells and flexible CF. Our suggested approach has the potential to change CMS optimization modeling by providing a novel and effective solution that has not yet been investigated.

The paper is structured as follows: The literature review section examines relevant literature on cell formation and machine cell layout using GA. In the methodology section, we introduce the proposed approach and performance measurement. The numerical example section presents a case study. In the results and discussion section, we analyze and discuss the findings of the study. Finally, the conclusion section concludes the paper by offering potential directions for future research and summarizing the main conclusions.

## Literature review

Designing a manufacturing cell is a challenging task that carries significant implications for an organization's operations. The process is highly complex and falls under the category of non-deterministic polynomial-time hard (NP-hard) problems, which means that the problem's complexity increases exponentially as the solution space expands. To overcome this challenge, many researchers have used efficient meta-heuristic approaches such as GA to solve the problem efficiently. In this study, we focus on two crucial steps in designing a manufacturing cell: CF and cell machine layout design. We review the survey work done in establishing a novel GA paired with GT to construct CMS, which has proven to be a highly effective approach to tackling complex issues in this domain.

### Cell formation by GA

Numerous studies have been conducted in both academic and industrial settings to apply GA to solving CF issues. One of the most notable works is the enhanced grouping genetic approach developed by Tunnukij and Hicks [12]. This approach did not specify the manufacturing cells as well as the number of parts and machines inside each cell. To address this limitation, Arkat et al. [13] proposed a bi-objective genetic approach to reduce the exceptional elements and voids inside machine cells. This helps to reduce intercellular movements and achieve an efficient solution in a reasonable amount of time. Another novel approach to CF problems with alternative part routes was introduced by Ozcelik and Saraç [14]. Their hybrid algorithm combined GA with a modified sub-gradient algorithm to reduce the weighted sum of exceptional elements and voids. These studies provide valuable insights and solutions for improving manufacturing operations through the use of GA in CF. Khaksar et al. [15] developed a novel integer linear programming model for optimizing the multi-floor layout design. The model employed GA that considered multiple objectives, including minimizing intracellular and intercellular costs, new machine purchases, and machine processing. Paydar and Saidi-Mehrabad [16] suggested a linear fractional programming model to maximize grouping efficacy for CF problems with an undetermined number of cells. The authors utilized a hybrid meta-heuristic algorithm that merges variable neighborhood search and GA to address this issue. These studies increase the state of the art in CMSs by proposing novel optimization models and novel algorithms for dealing with complex manufacturing issues. Saeidi et al. [17] introduced a mathematical model that addresses multiple objectives in the context of CF. They introduced a meta-heuristic algorithm based on GA and Fuzzy Goal Programming (FGP) approach to treat the model. The model takes into account various input parameters such as a sequence of operations, production volume, machine redundancy, cost of machines, alternative processing plans, and processing time. The authors convert the multi-objective model into a single objective by utilizing FGP method. Pachayappan and Panneerselvam [18] presented a hybrid GA aimed at improving both grouping efficiency and group efficacy performance measures. They evaluated their algorithm against four other existing algorithms by conducting a full factorial experiment. In this experiment, they treat the problem as one factor and the method as another, allowing for a comprehensive comparison of the

different algorithms. The results of this experiment demonstrate the superior performance of the developed hybrid GA. Taken together, the contributions of studies offer valuable insights into the development and optimization of CF processes. Imran et al. [19] offered a mathematical model to reduce the value-added Work-in-Process (WIP), and their approach was solved by integrating a hybrid GA and simulation. This new approach offered several advantages by utilizing the strengths of both techniques. A novel approach was tested in a local auto part supply company for CF, and the results showed that reducing waiting and throughput times had a positive impact on reducing the value-added WIP. Branco & Rocha [20] introduced a hybrid GA to address the CF issues. Their strategy entailed employing a greedy constructive method to optimize the utilization of machines within a cell while minimizing the need for motion between cells. The constructor technique and k-means algorithm were used to more precisely categorize the machine cells generated by the GA into part families. Sowmiya et al. [21] applied GA to address CF issues while considering an alternative processing route. A ranking index was utilized to calculate the correlation value, which was then used to produce parent chromosomes with the same number of genes as the number of parts. Their goal was to find the optimal processing route that would optimize the efficiency with which parts were grouped. The results of the suggested method were compared to those of the best machine-part CF found in the literature. The research showed that the proposed strategy outperformed the existing methodologies in 10% of the test cases. Hazarika & Laha [22] implemented GA to solve CF problems with multiple alternative processing routes, sequences of processes, and part volumes. Their objective was to minimize the total intercellular movements of parts based on the optimal alternative processing route. The results of the study, which was based on five benchmark problems, showed that the performance of the suggested strategy was either comparable to or better than that of the existing approaches in terms of the selection of the best path and the overall movement of parts within cells. Shashikumar et al. [23] utilized an integrated approach combining GA and membership index to address the CF problem. Their developed method aimed to optimize the grouping efficacy of the parts while minimizing the number of machine setups required. The study discovered that the suggested approach performed better than the current approaches in terms of reducing the number of machine setups while retaining good grouping efficacy.

## Machine cell layout by GA

The machine cell layout design problem in CMSs has been investigated by numerous authors. For instance, Chandrasekar and Venkumar [24] proposed a hierarchical GA approach to address the problems of CF and cell machine layout design. Their approach takes an incidence matrix of machine parts and a sequence of operations as input data and evaluates the quality of CMS design based on measures such as grouping efficacy and grouping efficiency. Javadi et al. [25] implemented a novel mathematical framework for the inter- and intra-cell layout problem in a dynamic environment. To mitigate the high costs associated with reorganization and cellular movements, an electromagnetism-like (EM-like) algorithm was proposed in conjunction with GA. The algorithms were assessed using various numerical examples segmented into three problem sizes, namely large, medium, and small. The results show that the hybrid EM-GA approach outperforms other methods. However, complex problems require efficient computational solutions, so Forghani and Mohammadi [26] developed an integrated approach that combines GA and takes into account various factors such as machine capacity, alternative process routings, demand for components, cell size, multi-row layout of machines within cells, and aisle lengths between machines. To enhance practicality, GA parameters were established using a design of experiments (DOE) approach, which determined the effects of individual

factors and any significant interactions among them on process performance. A comprehensive mathematical model was introduced by Deep and Singh [27] for enhancing dynamic part production by designing machine cells. This model provides flexibility in production planning, allowing for the production of various product mixes within the capacity limits of the manufacturing cell without the need for alterations to its configuration. To reduce overall costs, a heuristic based on GA was proposed to solve the model. The model takes into account several manufacturing factors, including production volume, subcontracted part operations, material handling, and machine capacity. Similarly, Feng et al. [28] introduced a comprehensive Mixed-Integer Linear Programming (MILP) model to solve the integrated CF and layout problem. The authors have introduced two hybrid approaches, Simulated Annealing Linear Programming (SALP) and Genetic Algorithm Linear Programming (GALP), to solve large-scale issues. The illustrative example's findings suggest an increase in machine utilization rate and a decrease in production costs. Overall, the proposed models can improve manufacturing processes' efficiency and offer cost-saving benefits. Maleki et al. [29] tackled the CF and machine layout problems by formulating them as multi-objective programming problems and solving them through GA. To improve on existing models, they introduced new objective functions by incorporating the Analytical Hierarchy Process (AHP) to account for multiple factors such as demand, capacity, and cost. These objectives aimed to maximize the significance of allocating parts and machines in the same cell. The proposed model provides the optimal machine location, process plan, and cell configuration for any part type during the planning horizon. Forghani and Ghomi [30] presented a mathematical model that integrates machine layout, CF, and cell scheduling issues in both conventional and virtual CMS. The model aims to minimize handling costs and cycle time while taking alternative processing routes into account. They employed three meta-heuristic algorithms, including GA, Memetic Algorithm (MA), and Simulated Annealing (SA), to solve the issues. The results showed that the proposed SA outperformed the GA and MA. Forghani et al. [31] proposed a hybrid solution approach that combines SA with GA to minimize material handling costs and electric energy consumption. A MILP was developed to deal with routing, assembly aspects, and energy consumption. The proposed approach shows promise in optimizing CF and layout design problems. Modrak et al. [32] used the Taguchi method to determine the best configuration of parameters that affect the performance of GA in layout design and CF problems. They investigated the impact of noise variables, including balance weight factor, probability of mutation, and probability of crossover, on the optimal combination of genetic operators. This study provides valuable insights into the design of an effective GA for solving CF and layout problems. Al-Zuheri et al. [33] introduced a new hybrid approach that combined GA and the desirability function to solve complex optimization problems related to CF and machine cell layouts. However, they did not consider that the measure of flexibility must be calculated for each value of the weight factor.

Some recently developed approaches from the related literature are summarized in Table 1. Several recent studies have also addressed the design considerations for CF and machine cell layouts, but they have not focused on the dynamic flexible layout within the CMS design to improve production flexibility. Despite the popularity of GA as an optimization approach for such problems, the static layout was dominant in the presented solutions. In this study, GA is employed to identify machine cells and part families based on Grouping Efficiency (GE) as a fitness function. In contrast to previous research, which considered grouping efficiency with a weight factor ($q = 0.5$), this study utilizes various weight factor values (0.1, 0.3, 0.7, 0.5, and 0.9), which will produce a variety of optimal solutions for each instance. In order to evaluate the optimal solution using the Multi-Criteria Decision-Making (MCDM) method, many MCDM models can be used in optimal selection, such as TOPSIS, Preference Ranking

**Table 1. Synopsis and comparison of the papers in the relevant literature.**

| References | CF | Cell layout | Objectives | | | | Parameters | | | | | |
|---|---|---|---|---|---|---|---|---|---|---|---|---|
| | | | O1 | O2 | O3 | O4 | P1 | P2 | P3 | P4 | P5 | P6 |
| Tunnukij and Hicks [12] | √ | | | | | | | | | | | |
| Khaksar et al. [15] | √ | √ | √ | | | | √ | √ | √ | √ | | |
| Saeidi et al. [17] | √ | | | | | | √ | √ | √ | √ | | |
| Pachayappan and Panneerselvam [18] | √ | | | √ | | | | | | | | |
| Chandrasekar & Venkumar [24] | √ | √ | | √ | | | √ | | | | | |
| Javadi et al. [25] | √ | √ | √ | | | | | | | | | |
| Forghani and Mohammadi [26] | √ | √ | | | | | √ | √ | √ | √ | | |
| Deep and Singh [27] | √ | √ | √ | | | | | | √ | √ | | |
| Maleki et al. [27] | √ | √ | √ | | | | √ | | | | | |
| Forghani et al. [31] | √ | √ | | | | | | √ | | | | |
| Al-Zuheri et al. [33] | √ | √ | √ | √ | | | √ | √ | √ | √ | √ | |
| Current study | √ | √ | √ | √ | √ | √ | √ | √ | √ | √ | √ | √ |

O1: Minimization of the cost of material handling between machines as well as inter- and intracellular movements; O2: Maximize group efficiency; O3: Maximize system utilization; O4: Maximize system flexibility; P1: Processing sequence; P2: Processing time; P3: Demand volume; P4: Machine capacity; P5: Max number of operations; P6: Varying weight factor ($q$).

Organization Method for Enrichment Evaluation (PROMETHEE), AHP, ELimination Et Choix Traduisant la REalite (ELECTRE), and so on [34]. The TOPSIS technique is used in this paper because it is easier to use, does not have strict rules about how the data is distributed or the size of the sample, and is better for sorting sample data internally [35], this paper adopts the TOPSIS technique. Moreover, the selection of the TOPSIS technique was based on its previous successful use in resolving decision-making problems of a similar nature [36]. Here, we aim to obtain a flexible layout design for CMS through the combination of hybrid GA and TOPSIS technique, which will provide an effective method for solving complex optimization problems related to CF and machine cell layouts.

## Methodology

The proposed approach is developed with the aim of designing a CMS layout that satisfies the flexibility needs in response to the dynamic manufacturing requirements. It is based on the principles of GA and TOPSIS. The structure of this methodology is illustrated in Fig 1.

### Genetic algorithm

GA is a well-liked meta-heuristic tool for solving difficult problems in the CMS. The underlying concept of GA is based on the principles of natural selection and genetics, which were initially proposed by Holland [37]. The following sequence of steps for GA can be applied to the setting of part-machine CF [32, 33].

The first step involves identifying the initial population of the chromosome. Machine groups are then paired with part families based on the results of the fitness function's evaluation of each population. The next phase is the creation of offspring by genetic operations like crossover and mutation. The goal is to increase the population variety to improve chromosomal fitness. A repair approach is then used after the genetic operation to determine each chromosome's fitness function value. To better understand the mechanism of the suggested GA, Fig 2 represents a flowchart.

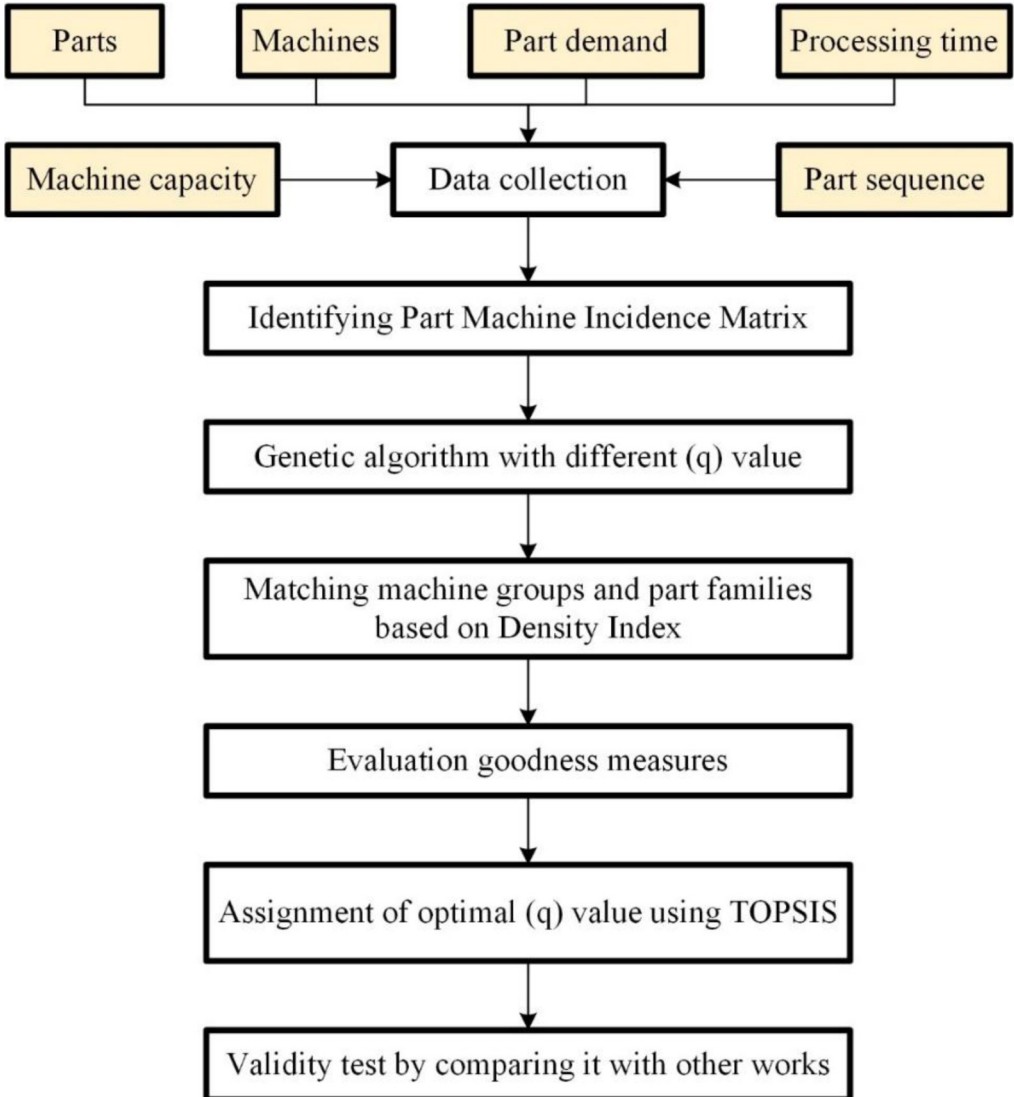

**Fig 1. Flowchart of the proposed methodology.**

Next, the best-performing chromosome is chosen when the entire evolutionary process is complete. That is accomplished by selecting the individual who scores highest on the fitness function. In order to apply GA to the CF problem, we must first determine the starting population, then assess the performance of each subpopulation, apply the appropriate genetic operations, fix the chromosome, and finally pick the chromosome with the best results. The proposed approach has the potential to address complex CMS problems and enhance the overall efficiency of the manufacturing process.

## Chromosome representation

The chromosome representation is crucial since it determines how the problem's variables are encoded. Fig 3 shows that chromosomes are commonly represented by integer numbers and classified as machine or part chromosomes. Each integer value represents a machine cell on machine chromosomes, and gene positions dictate machine order. The process's part family is

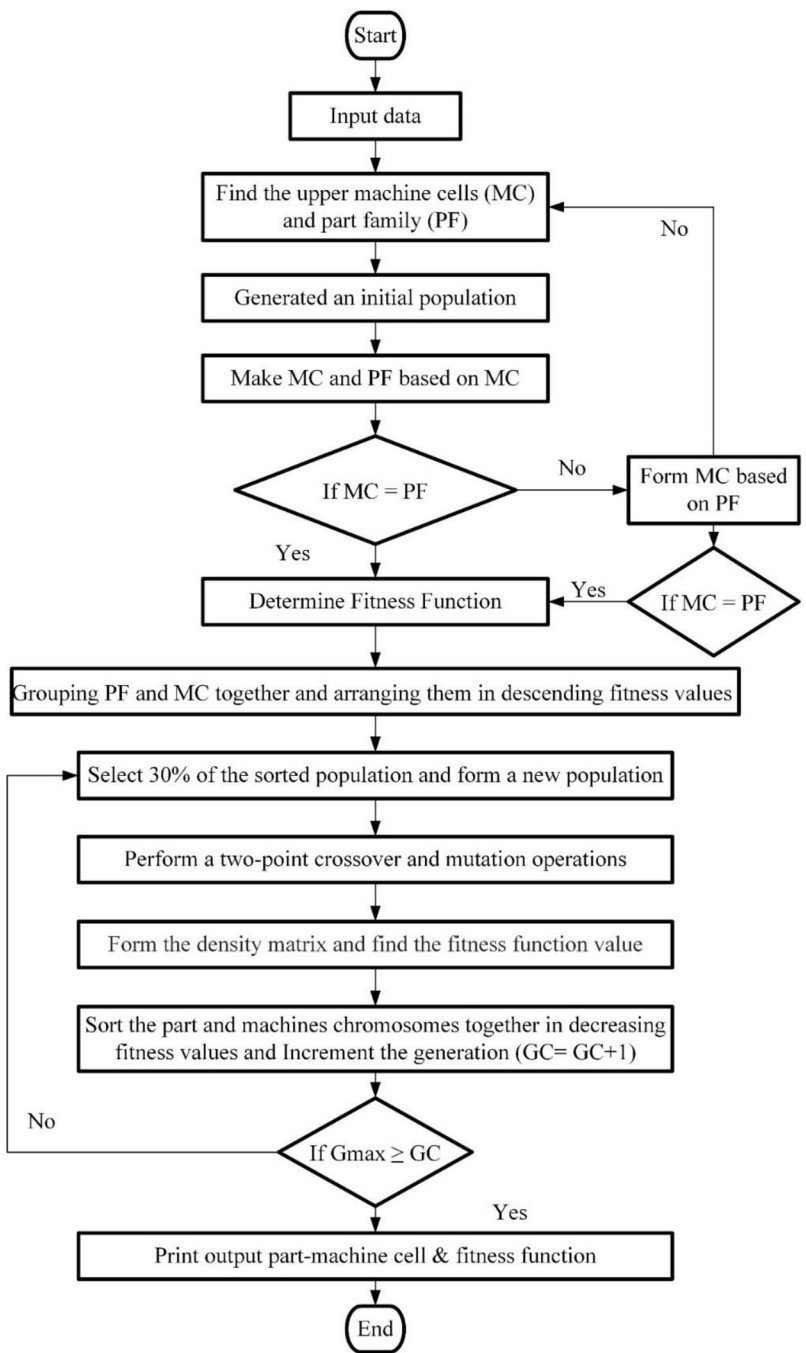

**Fig 2. Flowchart for proposed GA.**

represented by part chromosomes. The portion types' order depends on gene locations. This is represented by a gene value on the part or machine chromosome. This encoding lets the GA efficiently find the best solution by investigating machine and part assignments.

**Population initialization.** To address the part-machine CF problem, an initial population is generated at random and split into two sub-populations: the machine population, which

| Gen position | 1 | 2 | 3 | 4 | 5 | 6 | 7 | 8 |
|---|---|---|---|---|---|---|---|---|
| Part chromosome 1 | 1 | 1 | 2 | 3 | 3 | 2 | | |
| Part chromosome 2 | 2 | 1 | 2 | 1 | 1 | 3 | | |
| Machine chromosome 1 | 2 | 2 | 3 | 3 | 1 | 3 | 2 | 2 |
| Machine chromosome 2 | 2 | 1 | 3 | 1 | 2 | 2 | 3 | 3 |

**Fig 3. Part and machine chromosome representation.**

contains machine chromosomes, and the part population, which contains part chromosomes. It is crucial that the number of chromosomes in each population be equal.

**Fitness function.** The proposed method employs a fitness function that seeks to maximize grouping efficiency, a metric defined by the following formula [38]:

$$GE = q\,\eta_1 + (1-q)\,\eta_2. \tag{1}$$

Here, $\eta_1$ represents the ratio of "1" elements to the total number of elements within the diagonal blocks, while $\eta_2$ represents the ratio of "0" elements to the total number of elements outside the diagonal blocks. The parameter "$q$" allows for a weighting of the two ratios, with higher values of "$q$" prioritizing $\eta_1$ and lower values prioritizing $\eta_2$. The proposed approach seeks to maximize the density of "1" items within diagonal blocks while decreasing the density of "0" elements outside those blocks; therefore, this fitness function makes sense in this context. The use of GE as a fitness function has been shown to yield superior results compared to other metrics in previous studies [39, 40] making it a natural choice for the current work.

**Density index.** The density index is a metric used in manufacturing to pair machine cells with part families. The calculation of the density index involves associating a particular machine cell with a family of parts and computing the density index score. The score reflects how well the machine cell is matched to the part family based on the number of 1s in the part-machine cell matrix as well as the size of the part-machine cell matrix. The formula for the density index is [18]:

$$D_{ij} = \frac{\text{Number of 1s in the part} - \text{machine cell}}{\text{Size of the part} - \text{machine cell}}, \tag{2}$$

where $i$ is the index of the machine cell, and $j$ is the index of the part family.

**Genetic operators.** Crossover and mutation are two fundamental genetic operators that produce a new set of chromosomes or "offspring" that are employed to produce the following generation. In crossover, genetic material from two parents is combined to produce an organism with traits from both. The two-point crossover method, as depicted in Fig 4, is used in this study. This method entails choosing two crossing spots and exchanging genetic material between them. Conversely, mutation creates new forms of diversity in a population by arbitrarily changing genes on chromosomes. In our implementation, each offspring produced by the crossover undergoes a mutation, which involves the random swap of genes at two

| Gen position | 1 | 2 | 3 | 4 | 5 | 6 |
|---|---|---|---|---|---|---|
| Part chromosome 1 | 1 | 1 | 3 | 1 | 2 | 2 |
| Part chromosome 2 | 2 | 1 | 2 | 3 | 1 | 2 |
| Offspring 1 | 1 | 1 | 2 | 3 | 2 | 2 |
| Offspring 2 | 2 | 1 | 3 | 1 | 1 | 2 |

**Fig 4. Two-point crossover method.**

randomly chosen positions. Together, crossover and mutation create a diverse population that can explore the search space and converge on an optimal solution.

**Repair strategy.** During optimization, a chromosome that is poorly organized or impossible to implement could be produced. This happens when a cell is not properly assigned to a machine or a part, making the chromosome useless. A random gene is picked from both the machine and the part chromosomes to solve this problem. To keep the chromosome well-structured and practical, the selected gene is given the missing value if other genes on the same chromosome already have that value.

**Parameters setting.** To improve the effectiveness of our hybrid GA and achieve optimal outcomes, we have identified four key parameters that require meticulous calibration: Crossover probability ($Pc$), Mutation probability ($Pm$), Maximum number of Iteration ($MaxIt$), and Population size ($Npop$). We have utilized the Taguchi method as a Design of Experiment (DOE) technique to establish suitable values for each parameter based on an exhaustive analysis of the algorithm. This has involved categorizing each parameter into three distinct levels, as demonstrated in Table 2.

## TOPSIS

In order to guarantee that the generated CMS design layout from GA runs possesses high cell adaptability, it is advisable to conduct a comprehensive assessment of the impact of varying weight factor values ($q$) on the system's performance metrics. To achieve this, the TOPSIS method is utilized to evaluate the quality of the solution [41]. Hwang and Yoon [42] proposed TOPSIS as a method for selecting the best alternative based on the concept of the compromise

**Table 2. Design parameters and their levels.**

| Factor | Level 1 | Level 2 | Level 3 |
|---|---|---|---|
| $Pc$ | 0.2 | 0.5 | 0.8 |
| $Pm$ | 0.1 | 0.2 | 0.3 |
| $MaxIt$ | 100 | 250 | 500 |
| $Npop$ | 40 | 60 | 80 |

solution. The selection of an alternative that is closest to the ideal solution while being farthest from the negative ideal solution defines a compromise solution [43]. The TOPSIS algorithm consists of six steps and is executed using a priority matrix ($X_{ij}$), where $A = \{Ai|i = 1,\ldots,n\}$ denotes the alternatives and $C = \{Cj|j = 1,\ldots,m\}$ denotes the criteria, with corresponding weights ($w_j$). The procedure can be outlined as follows [35, 36, 44]:

**Step 1.** It is required to normalize ratings collected from several sources or on various scales to ensure a fair comparison across various entities, such as products or services. In this study, we take a two-stage procedure for determining standard scores. At first, we determine the mean and standard deviation of each entity's ratings across all sources. Second, we adjust the original ratings for each source by subtracting the average rating and dividing it by the standard deviation of the entity's ratings. Therefore, Eq (3) is given as follows:

$$r_{ij} = \frac{X_{ij}}{\sqrt{\sum_{i=1}^{n} X_{ij}^2}}, (i = 1, \ldots, n; j = 1, \ldots, m). \tag{3}$$

**Step 2.** The relative relevance of each rating category is used to create weighted normalized ratings. This will guarantee that the final grade fairly reflects the performance. More specifically, a relative value is assigned to each factor. The next step is to multiply each category's rating by its weight. The weighted average is scaled to 0–1, with 1 being the highest rating. By using this method, we can evaluate performance fairly and accurately. In this regard, we have

$$v_{ij} = w_j \times r_{ij} (i = 1, \ldots, n; j = 1, \ldots, m). \tag{4}$$

**Step 3.** Computing the Positive Ideal Points (PIS) and negative Ideal Points (NIS) is fundamental to multi-criteria decision-making. These two factors play an important role in identifying the best and worst possible decisions. The PIS is the best solution that maximizes all criteria, whereas the NIS is the worst solution that minimizes all. These two points allow decision-makers to calculate the distance between the present solution and the optimal solution and make informed decisions to improve the outcome. Thus, calculating the PIS and NIS is essential in multi-criteria decision-making for efficient and effective decision-making. We employ the following formula to measure the PIS and NIS.

$$PIS = \{v_1^+, \ldots \ldots, v_c^+\}, v_j^+ = \left\{ \max\left(v_{ij}\right) \text{ if } j \in J_1; \min\left(v_{ij}\right) \text{ if } j \in J_2 \right\}, \tag{5}$$

$$NIS = \{v_1^-, \ldots \ldots, v_c^-\}, v_j^- = \left\{ \min\left(v_{ij}\right) \text{ if } j \in J_1; \max\left(v_{ij}\right) \text{ if } j \in J_2 \right\}, \tag{6}$$

**Step 4.** Separation measures are used to evaluate the distance between each alternative and the ideal solutions for the decision criteria. The two sorts of separation measurements are positive and negative. Positive Separation ($PS_i^+$) measures how closely an alternative matches the NIS, the worst possible performance for each criterion. A $PS_i^+$ value of 0 suggests the alternative is farthest from the NIS, while 1 implies it is closest. Alternatives with greater $PS_i^+$ values are better. In contrast, Negative Separation ($NS_i^-$) measures how near an option is to the PIS, the best performance for each criterion. The $NS_i^-$ value varies from 0 to 1, with 0 indicating the furthest distance from the PIS and 1 indicating the greatest proximity. Higher $NS_i^-$

values mean inferior alternatives. $PS_i^+$ and $NS_i^-$ are measured using these formulas:

$$PS_i^+ = \sqrt{\sum_{j=1}^{c} \left(v_j^+ - v_{ij}\right)^2}, \tag{7}$$

$$NS_i^- = \sqrt{\sum_{j=1}^{c} \left(v_j^- - v_{ij}\right)^2}. \tag{8}$$

**Step 5.** The relative closeness is determined for each alternative. This is done by calculating the following formula:

$$C_i^+ = \frac{NS_i^-}{NS_i^- + PS_i^+}. \tag{9}$$

The Closeness to the PIS value can take on values between 0 and 1, with 0 denoting the farthest removed from the optimal solution and 1 the closest. The closer an option is to PIS, the better it is.

**Step 6.** Finally, we evaluate the alternatives according to how similar they are to the ideal answer, with the highest value representing the best option. With TOPSIS, the best option is the one that is closest to the ideal solution, and the ranking of alternatives is determined by how well they perform in relation to the decision criteria. The alternatives are ranked in TOPSIS depending on how well they perform in comparison to the other options, as TOPSIS is a relative ranking approach. Therefore, if all of the alternatives are performing poorly, the ranking may not reflect the absolute performance of an alternative. If there is no discernible difference in performance between the options being considered, the rating is meaningless.

## Measure of performance

Numerous researchers have devised a variety of efficiency measures. This measure can be employed to assess the excellence of CF. The effectiveness of block diagonalization is evaluated by analyzing the resultant matrix's quality. The current study will focus on three different types of performance measures. The cost of intracellular and intercellular movement (TC), system utilization (SU), and system flexibility (SF). These three performance measures are computed using Eqs (10), (14) and (19), respectively [45–48]:

$$TC = C_1 N_i + C_2 \sum_{j=1}^{m} d_{ij} k_{ij} \tag{10}$$

Such that

$$\begin{cases} d_{ij} \text{ for stright line} = \dfrac{N+1}{3}, \\ d_{ij} \text{ for square layout} = 2\dfrac{\sqrt{N}}{3}, \end{cases} \tag{11}$$

where $TC$ is the total cost of intracellular and intercellular movement ($TC$) for the $i$th configuration, $d_{ij}$ stands for the expected distance moved between two machines, $k_{ij}$ is the number of moves between two machines by each part for the $i$th configuration, $N_i$ represents the number of inter-cell movements for $i$th configuration, $N$ is a number of machines in the group, and finally, $C_1$ stands for the cost of an inter-cell movement and $C_2$ shows the cost per unit distance

of an intracell movement. $j$ is the cell index ($j = 1, \ldots, m$).

$$MU_{jc}(t) = \frac{\sum_{i=1}^{n_c} t_{ijc} D_i(t)}{C_{jc}(t)}, \tag{12}$$

$$CU_C(t) = \frac{1}{m_c} \sum_{j=1}^{m_c} \left[ \frac{\sum_{j=1}^{m_c} t_{ijc} D_i(t)}{C_{jc}(t)} \right], \tag{13}$$

$$SU_{jc}(t) = \frac{1}{C(t)} \sum_{c=1}^{C(t)} \frac{1}{m_c} \sum_{i=1}^{m_c} \left[ \frac{\sum_{k=1}^{m_c} t_{ijc} D_i(t)}{C_{jc}(t)} \right], \tag{14}$$

where $MU_{jc}(t)$ machine utilization, type $j$ in cell $C$ at the time, $t_{ijc}$ is processing time of part $i$ on machine $j$ in cell $c$, $D_i(t)$ demand of part at time $t$, $C_{jc}(t)$ represents the capacity of machine, type $j$ in cell $c$ at time $t$. Furthermore, $n_c$ refers to the number of parts in cell $c$, $CU_c(t)$ is the utilization of the cell at time $t$, $m_c$ represents the number of machines inside cell $c$, $SUjc$ is the utilization of the overall manufacturing system at time $t$, and $C(t)$ displayed the number of manufacturing cell at time $t$.

$$MF_{jc}(t) = \sum_{o=1,i=1}^{n_{jo}} \frac{SMC_{jc}(t)}{C_{jc}(t)} \times \frac{SMF_{jc}(t)}{N_{O_{jc\ max}}} \tag{15}$$

$$SMC_{jc}(t) = C_{jc}(t) - \sum_{i=1}^{m_c} t_{ijc} D_i(t), \tag{16}$$

$$SMF_{jc}(t) = N_{O_{jc\ max}} - n_{io}, \tag{17}$$

$$CF_{jc}(t) = \frac{1}{m_c} \sum_{j=1}^{m_c} \sum_{o=1,i=1}^{n_{jo}} \frac{SMC_{jc}(t)}{C_{jc}(t)} \times \frac{SMF_{jc}(t)}{N_{O_{jc\ max}}}, \tag{18}$$

$$SF_{jc}(t) = \frac{1}{C(t)} \sum_{c=1}^{C(t)} \frac{1}{m_c} \sum_{j=1}^{m_c} \sum_{o=1,i=1}^{n_{oj}} \frac{SMC_{jc}(t)}{C_{jc}(t)} \times \frac{SMF_{jc}(t)}{N_{O_{jcmax(t)}}}, \tag{19}$$

where $MFjc(t)$ represents the flexibility of machine type $j$ in manufacturing system $c$ at time $t$, $n_{oj}$ refers to the number of operations performed on machine type $j$, $SMCjc(t)$ shows the slack in the capacity of machine type $j$ at time $t$, $SMFjc(t)$ displays the slack in the capability of machine type $j$ at time $t$, $N_{O_{jc\ max(t)}}$ is the maximum number of operations that can be performed on machine type $j$ in cell $c$, $CFjc(t)$ represents the flexibility of the cell at time $t$, $SFjc(t)$ stands for the flexibility of the overall manufacturing system at time $t$.

## Numerical example

In order to demonstrate the efficacy of the proposed meta-heuristic. The Hydraulic Industries State Company in Baghdad-Iraq is used as a real-life case study. This company is an engineering firm focused on four main areas: compressed air, hydraulics, lubrication, and manufacturing service equipment [49]. It has four production plants that manufacture different products, such as dampers, hydraulic and pneumatic components, electrical equipment, and specialized service equipment. The damper manufacturing line was selected based on the machines' positioning in the factory's functional layout, which promotes more intercellular movement

between units. There are two sections in the damper factory: the processing section and the assembly section. These departments are the following: (welding department presses department and polishing department). There are 27 parts in the finished product (the damper). Eight parts are manufactured once the final parts that are purchased off the market are eliminated. The data for 8 parts on 22 machines is presented in Table 3, including the times, machine operating sequence, and the monthly volume of the products. Moreover, the machine capacity for each machine is 37.5 hours per week.

Table 4 represents: If the part is processed in the machine, the binary part-machine incidence matrix is 1; otherwise, it is 0. The last row of Table 4 represents the maximum number of operations for each machine. The grouping efficiency of the existing manufacturing system is based on Eq (1) as given below:

$$GE = 0.5\left(\frac{38}{176}\right) + (1 - 0.5)\frac{0}{0} = 10.79\%.$$

While SU and SF of the current manufacturing system are based on Eqs (10)–(17) as given in Table 5.

## Results and discussion

The GA was created using the MATLAB R2017b software on a personal computer equipped with an Intel Core i5 CPU with a speed of 2.4 GHz and 8 GB of RAM. The GA code is available at (https://zenodo.org/records/10147377). The Taguchi method creates a matrix experiment that includes nine trials with four factors and three levels using the orthogonal array in Table 6. The responses are assigned to their respective trials. Subsequently, the Taguchi method evaluates the experiment and produces a visual representation of the signal-to-noise ratio of each factor in Fig 5. The purpose of the figure is to compare the different levels for each factor and select the optimal one. The analysis of the signal-to-noise ratio reveals that, among all levels, level 1 is preferable for both crossover probability and mutation probability, while level 3 is preferable for both *MaxIt* and *Npop*. After establishing the parameters for the GA, the weight factor ($q$) was varied for further experimentation. Specifically, weight factors of 0.1, 0.3, 0.5, 0.7, and 0.9 were used, and the results were compared. The findings from these experiments are summarized in Table 7. This research identified six decision variables: group efficiency, the total cost of intracellular and intercellular movements, system utilization, system flexibility, exceptional elements, and voids. We assign weight to decision variables. The decision variable values corresponding to each weight factor value are displayed in Table 8.

**Table 3. Product information.**

| Part Type | Part Code | Processing Time(s) | Machine Sequence | Demand volume |
|---|---|---|---|---|
| Internal cylinder | P1 | 45–30 | 14–16 | 4200 |
| Inner Pipe | P2 | 10-35-20-15-30-25 | 13-12-9-10-17-6 | 4200 |
| Body | P3 | 25-26-120-15-120-20 | 14-16-20-4-15-2 | 4200 |
| Ring | P4 | 10-30-45-15-17-20 | 13-17-6-3-1-2 | 8400 |
| Reservoir Head | P5 | 10-10-17-25 | 13-7-5-6 | 4200 |
| Shaft | P6 | 30-55-17-20-15-120-120-30-60 | 14-12-11-9-10-21-22-19-8 | 4200 |
| Internal Washer | P7 | 10-15-45 | 13-7-18 | 4200 |
| Plate | P8 | 30–35 | 14–12 | 4200 |

**Table 4. Part-machine incidence matrix.**

| P | \ | | | | | | | | | | | | Machines | | | | | | | | | |
|---|---|---|---|---|---|---|---|---|---|---|---|---|---|---|---|---|---|---|---|---|---|---|
| | 1 | 2 | 3 | 4 | 5 | 6 | 7 | 8 | 9 | 10 | 11 | 12 | 13 | 14 | 15 | 16 | 17 | 18 | 19 | 20 | 21 | 22 |
| 1 | 0 | 0 | 0 | 0 | 0 | 0 | 0 | 0 | 0 | 0 | 0 | 0 | 0 | 1 | 0 | 1 | 0 | 0 | 0 | 0 | 0 | 0 |
| 2 | 0 | 0 | 0 | 0 | 0 | 1 | 0 | 0 | 1 | 1 | 0 | 1 | 1 | 0 | 0 | 0 | 1 | 0 | 0 | 0 | 0 | 0 |
| 3 | 0 | 1 | 0 | 1 | 0 | 0 | 0 | 0 | 0 | 0 | 0 | 0 | 0 | 1 | 1 | 1 | 0 | 0 | 0 | 1 | 0 | 0 |
| 4 | 1 | 1 | 1 | 0 | 0 | 1 | 0 | 0 | 0 | 0 | 0 | 0 | 1 | 0 | 0 | 0 | 1 | 0 | 0 | 0 | 0 | 0 |
| 5 | 0 | 0 | 0 | 0 | 1 | 1 | 1 | 0 | 0 | 0 | 0 | 0 | 1 | 0 | 0 | 0 | 0 | 0 | 0 | 0 | 0 | 0 |
| 6 | 0 | 0 | 0 | 0 | 0 | 0 | 0 | 1 | 1 | 1 | 1 | 1 | 0 | 1 | 0 | 0 | 0 | 0 | 1 | 0 | 1 | 1 |
| 7 | 0 | 0 | 0 | 0 | 0 | 0 | 1 | 0 | 0 | 0 | 0 | 0 | 1 | 0 | 0 | 0 | 0 | 1 | 0 | 0 | 0 | 0 |
| 8 | 0 | 0 | 0 | 0 | 0 | 0 | 0 | 0 | 0 | 0 | 0 | 1 | 0 | 1 | 0 | 0 | 0 | 0 | 0 | 0 | 0 | 0 |
| | 1 | 2 | 1 | 1 | 7 | 1 | 7 | 7 | 1 | 1 | 13 | 25 | 1 | 1 | 5 | 3 | 25 | 1 | 1 | 1 | 1 | 1 |

TOPSIS was calculated for all the decision variables to find the most optimal solution for forming and arranging the manufacturing cell. Table 9 shows the normalized decision matrix using normalized ratings by Eq (3). Afterwards, the normalized decision matrix is multiplied with the weight connected to each criterion utilizing Eq (4) as shown in Table 10. Then, we determine the PIS and NIS by using Eqs (5) and (6). Finally, we calculate the separation from the PIS and the NIS between alternatives, and the relative closeness to the ideal solution by using Eqs (7)–(9). According to Table 11, the optimal cellular manufacturing layout is achieved at a weight factor ($q$) of 0.3. Fig 6 shows the convergence of the algorithm at a weight factor of 0.3. Consequently, the layout of machine cells obtained with this q value was superior

**Table 5. Performance measures for the existing system.**

| Machine No. | Machines | | Cells | | System | |
|---|---|---|---|---|---|---|
| | Utilization | Flexibility | Utilization | Flexibility | Utilization | Flexibility |
| 1 | 0.2405 | 0 | 0.4767 | 0.1548 | 0.4767 | 0.1548 |
| 2 | 0.4244 | 0 | | | | |
| 3 | 0.2122 | 0 | | | | |
| 4 | 0.1061 | 0 | | | | |
| 5 | 0.1202 | 0.7541 | | | | |
| 6 | 0.9903 | 0 | | | | |
| 7 | 0.1768 | 0.5879 | | | | |
| 8 | 0.4244 | 0.4933 | | | | |
| 9 | 0.2829 | 0 | | | | |
| 10 | 0.2122 | 0 | | | | |
| 11 | 0.1203 | 0.8121 | | | | |
| 12 | 0.8842 | 0.1018 | | | | |
| 13 | 0.3537 | 0 | | | | |
| 14 | 0.9196 | 0 | | | | |
| 15 | 0.8488 | 0.1209 | | | | |
| 16 | 0.3961 | 0.2013 | | | | |
| 17 | 0.6367 | 0.3343 | | | | |
| 18 | 0.3181 | 0 | | | | |
| 19 | 0.2122 | 0 | | | | |
| 20 | 0.8489 | 0 | | | | |
| 21 | 0.8489 | 0 | | | | |
| 22 | 0.8489 | 0 | | | | |

**Table 6. Experiments of the Taguchi design.**

| Experiment No. | Pc | Pm | MaxIt | Npop | Pc | Pm | MaxIt | Npop | Fitness function |
|---|---|---|---|---|---|---|---|---|---|
| 1 | 1 | 1 | 1 | 1 | 0.2 | 0.1 | 100 | 40 | 0.7504 |
| 2 | 1 | 2 | 2 | 2 | 0.2 | 0.2 | 250 | 60 | 0.7408 |
| 3 | 1 | 3 | 3 | 3 | 0.2 | 0.3 | 500 | 80 | 0.7569 |
| 4 | 2 | 1 | 2 | 3 | 0.5 | 0.1 | 250 | 80 | 0.7504 |
| 5 | 2 | 2 | 1 | 1 | 0.5 | 0.2 | 500 | 40 | 0.7317 |
| 6 | 2 | 3 | 3 | 2 | 0.5 | 0.3 | 100 | 60 | 0.7236 |
| 7 | 3 | 1 | 3 | 2 | 0.8 | 0.1 | 500 | 60 | 0.7555 |
| 8 | 3 | 2 | 1 | 3 | 0.8 | 0.2 | 100 | 80 | 0.7345 |
| 9 | 3 | 3 | 2 | 1 | 0.8 | 0.3 | 250 | 40 | 0.7376 |

to the other layouts. At this q value, three cells were formed, as shown in Table 12. This layout provides an optimized solution for forming and arranging the manufacturing cell that maximizes group efficiency, minimizes intracellular and intercellular movement costs, maximizes system utilization, and maximizes system flexibility. In Table 12 the parts (P2, P4, 66, and P8) are considered exceptional parts because of processed in more than one machine cell. exceptional parts can be eliminated by duplicating the bottleneck machine, redesigning parts, eliminating the bottleneck operations, and releasing the capacity of the bottleneck machine. Overall, the results of this study demonstrate that the proposed heuristic using TOPSIS can effectively identify the most optimal solution for forming and arranging the manufacturing cell based on performance metrics. The methodology presented in this study can be useful in other manufacturing settings to optimize production processes and improve overall efficiency.

To evaluate the performance of the proposed approach, it was compared to previously studied techniques, including a hybrid GA [18] based on GE and MBO-FF [50] based on grouping efficacy [51]. Results from applying the proposed approach to 10 benchmark problems were analyzed and compared to those obtained using other approaches. Table 13 displays the outcomes of the performance assessments. The findings indicate that the proposed approach outperforms or achieves results that are on par with those presented in the literature, which

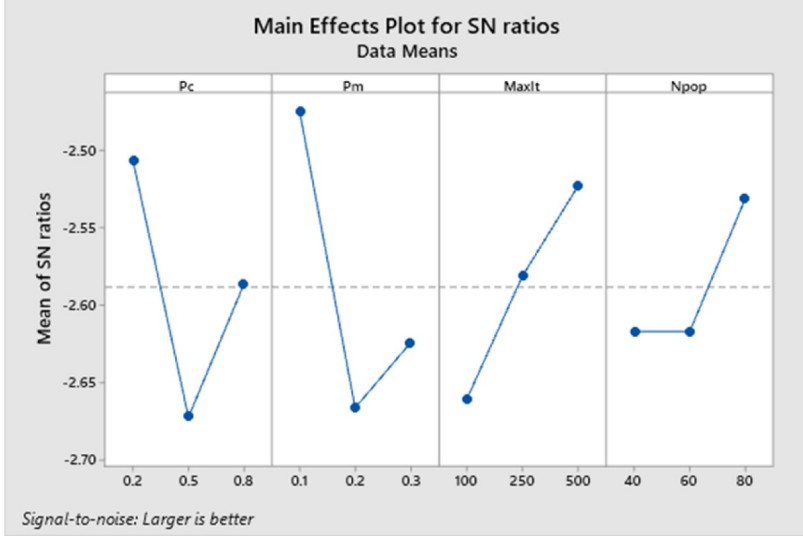

**Fig 5. The diagram of the Taguchi experiment.**

**Table 7. Machine and part chromosomes with varying weight factor values.**

| No. | q | Machine chromosome | Part chromosome | Fitness Function |
|---|---|---|---|---|
| 1 | 0.1 | 1 1 1 1 1 1 1 1 1 1 1 1 1 1 1 1 1 1 1 1 1 1 1 1 | 1 1 1 1 1 1 1 1 | 0.0216 |
| 2 | 0.3 | 3 1 3 1 3 3 3 2 2 2 2 2 3 1 1 1 3 3 2 1 2 2 | 1 2 1 3 3 2 3 1 | 0.7902 |
| 3 | 0.5 | 4 4 4 4 4 3 1 2 2 2 2 4 3 1 4 4 3 2 2 4 2 2 | 1 3 4 3 3 2 1 1 | 0.8073 |
| 4 | 0.7 | 4 4 4 4 4 3 1 4 1 4 2 2 3 2 4 4 3 1 1 4 1 2 | 2 3 4 3 3 2 1 2 | 0.6935 |
| 5 | 0.9 | 3 2 3 2 4 1 4 3 3 3 3 1 4 2 2 2 3 4 3 2 3 3 | 2 3 2 3 4 3 4 1 | 0.5767 |

**Table 8. Alternatives and decision variables with varying weight factor values.**

| q | GE | SU | SF | NV | EE | TC ($) |
|---|---|---|---|---|---|---|
| q = 0.1 | 0.0216 | 0.4764 | 0.1548 | 138 | 0 | 230 |
| q = 0.3 | 0.7902 | 0.4836 | 0.1464 | 26 | 6 | 84 |
| q = 0.5 | 0.8073 | 0.5176 | 0.1749 | 9 | 15 | 69.6703 |
| q = 0.7 | 0.6935 | 0.5226 | 0.1536 | 13 | 16 | 70.6732 |
| q = 0.9 | 0.5767 | 0.5435 | 0.1513 | 24 | 10 | 76.3455 |
| $W_j$ | 0.2000 | 0.2000 | 0.2000 | 0.1000 | 0.1000 | 0.2000 |

**Table 9. Normalized decision matrix.**

| q | GE | SU | SF | NV | EE | TC ($) |
|---|---|---|---|---|---|---|
| q = 0.1 | 0.0147 | 0.4182 | 0.4425 | 0.9627 | 0 | 0.8363 |
| q = 0.3 | 0.5654 | 0.4245 | 0.4185 | 0.1813 | 0.2415 | 0.3054 |
| q = 0.5 | 0.5500 | 0.4544 | 0.4998 | 0.0627 | 0.6038 | 0.2533 |
| q = 0.7 | 0.4724 | 0.4588 | 0.4390 | 0.0906 | 0.6441 | 0.2569 |
| q = 0.9 | 0.3927 | 0.4771 | 0.4316 | 0.1674 | 0.4025 | 0.2775 |

**Table 10. Weighted normalized decision matrix.**

| q | GE | SU | SF | NV | EE | TC ($) |
|---|---|---|---|---|---|---|
| q = 0.1 | 0.0029 | 0.0836 | 0.0885 | 0.0962 | 0 | 0.1672 |
| q = 0.3 | 0.1130 | 0.0849 | 0.0837 | 0.0181 | 0.0241 | 0.0610 |
| q = 0.5 | 0.1100 | 0.0908 | 0.0999 | 0.0062 | 0.0603 | 0.0506 |
| q = 0.7 | 0.0944 | 0.0917 | 0.0878 | 0.0090 | 0.0644 | 0.0513 |
| q = 0.9 | 0.0785 | 0.0954 | 0.0863 | 0.0167 | 0.0402 | 0.0555 |
| PIS | 0.1130 | 0.0954 | 0.0999 | 0.0062 | 0 | 0.0506 |
| NIS | 0.0029 | 0.0836 | 0.0837 | 0.0962 | 0.0644 | 0.1672 |

**Table 11. Final ranking for values of q.**

| q | $PS_i^+$ | $NS_i^-$ | $C_i^+$ | Rank |
|---|---|---|---|---|
| q = 0.1 | 0.1846 | 0.0645 | 0.2591 | 5 |
| q = 0.3 | 0.0347 | 0.1764 | 0.8354 | 1 |
| q = 0.5 | 0.0606 | 0.2224 | 0.7858 | 2 |
| q = 0.7 | 0.0682 | 0.1832 | 0.7285 | 4 |
| q = 0.9 | 0.0559 | 0.1589 | 0.7395 | 3 |

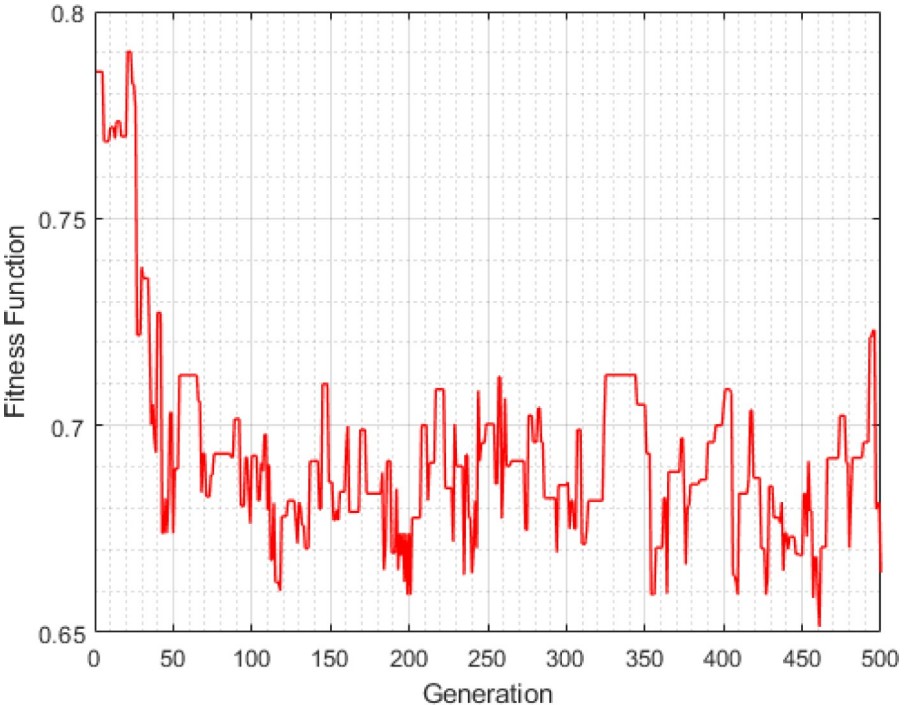

**Fig 6. Fitness function curve at weight factor 0.3.**

indicates the sensitivity of the proposed approach when taking a different value for the weighting factor to maximize grouping efficiency and grouping efficacy.

## Conclusion

In summary, this paper proposed a novel hybrid GA approach for optimizing the design of CF and machine layouts in CMS. Unlike traditional approaches that use a static layout and a fixed weight factor for machine utilization and cell movement, our approach incorporated the TOPSIS technique and allowed for flexible reconfiguration of the layout with a weight factor that can be adjusted to meet specific design goals. The use of a matrix-based chromosome structure in the GA led to an almost ideal solution design. The proposed approach considered various decision variables, including group efficiency, total cost of movements, system utilization, flexibility, exceptional elements, and voids. The efficacy of the developed approach was demonstrated through its successful application in resolving a real-life case study at the Hydraulic

**Table 12. Best part-machine incidence matrix.**

| P | 2 | 4 | 14 | 15 | 20 | 16 | 8 | 9 | 10 | 11 | 12 | 19 | 21 | 22 | 1 | 3 | 5 | 6 | 7 | 13 | 17 | 18 |
|---|---|---|----|----|----|----|---|---|----|----|----|----|----|----|---|---|---|---|---|----|----|----|
| 1 | 0 | 0 | 1 | 0 | 0 | 1 | | | | | | | | | | | | | | | | |
| 3 | 1 | 1 | 1 | 1 | 1 | 1 | | | | | | | | | | | | | | | | |
| 8 | 0 | 0 | 1 | 0 | 0 | 0 | | | | | 1 | | | | | | | | | | | |
| 2 | | | | | | | 0 | 1 | 1 | 0 | 1 | 0 | 0 | 0 | | | | 1 | | 1 | 1 | |
| 6 | | | 1 | | | | 1 | 1 | 1 | 1 | 1 | 1 | 1 | 1 | | | | | | | | |
| 4 | 1 | | | | | | | | | | | | | | 1 | 1 | 0 | 1 | 0 | 1 | 1 | 0 |
| 5 | | | | | | | | | | | | | | | 0 | 0 | 1 | 1 | 1 | 1 | 0 | 0 |
| 7 | | | | | | | | | | | | | | | 0 | 0 | 0 | 0 | 1 | 1 | 0 | 1 |

**Table 13. Comparison results.**

| Problem No. | Reference | Size | Grouping efficacy | | Grouping efficiency | |
|---|---|---|---|---|---|---|
| | | | MBO-FF | Our HGA | HGA [18] | Our HGA |
| 1 | King and Nakornchai [52] | 5 × 7 | 75 | 77.78 | 91.30 | 93.23 |
| 2 | Waghodekar and Sahu [53] | 5 × 7 | 69.57 | 78 | 82.61 | 88.23 |
| 3 | Kusiak and Cho [54] | 6 × 8 | 76.9 | 76.9 | NA | 87.5 |
| 4 | Kusiak and Chow [55] | 7 × 11 | 53.13 | 54.8 | NA | 78.23 |
| 5 | Chandrasekharan and Rajagopalan [40] | 8 × 20 | 85.25 | 85.25 | 95.83 | 95.83 |
| 6 | Stanfel [56] | 14 × 24 | 71.83 | 71.83 | 91.32 | 91.32 |
| 7 | Chan and Milner [57] | 15 × 10 | 92 | 92 | 96 | 96 |
| 8 | Nambirajan [58] | 20 × 40 | 59 | 59 | 82.99 | 85.8 |
| 9 | Chandrasekharan and Rajagopalan [40] | 20 × 40 | 100 | 100 | 100 | 100 |
| 10 | Stanfel [56] | 30 × 50 | 54.81 | 54.81 | 81.62 | 81.62 |

NA: Not Available

Industries State Company in Baghdad, Iraq, with a $q$ value of 0.3 resulting in the best cellular manufacturing layout. Moreover, the effectiveness of the suggested method was evaluated on ten standard problems, and its outcomes were examined and contrasted with those of other methods.

It is possible that the study's performance measurements are insufficient to fully capture the full scope of a manufacturing system's efficiency and productivity. Furthermore, reliability or machine breakdowns are not dealt with in this paper because all the machines are assumed to be in perfect condition. These are the limitations of this study. Based on the results of this research, this approach can be applied to a wider range of CMS design issues, including but not limited to reliability, group scheduling, production planning, and alternative layouts (such as U-shaped, double-row, or others). Moreover, future research can focus on integrating environmental considerations into the design and optimization of CMSs. This can involve incorporating criteria related to energy consumption, waste reduction, and carbon footprint in the decision-making process, contributing to the development of environmentally conscious manufacturing practices.

## Author Contributions

**Conceptualization:** Sawsan S. A. Al-Zubaidi, Luma A. H. Al-Kindi.

**Data curation:** Dhulfiqar Hakeem Dhayef.

**Formal analysis:** Dhulfiqar Hakeem Dhayef.

**Investigation:** Sawsan S. A. Al-Zubaidi, Luma A. H. Al-Kindi.

**Methodology:** Dhulfiqar Hakeem Dhayef.

**Resources:** Dhulfiqar Hakeem Dhayef.

**Software:** Dhulfiqar Hakeem Dhayef.

**Supervision:** Erfan Babaee Tirkolaee.

**Validation:** Sawsan S. A. Al-Zubaidi, Luma A. H. Al-Kindi, Erfan Babaee Tirkolaee.

**Writing – original draft:** Dhulfiqar Hakeem Dhayef.

**Writing – review & editing:** Erfan Babaee Tirkolaee.

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
