## [Decision Letter · Decision Letter 0]

8 Nov 2023

PONE-D-23-26796Cell Formation and Layout Design using Genetic Algorithm and TOPSIS: A Case Study of Hydraulic Industries State CompanyPLOS ONE

Dear Dr. Dhayef,

Thank you for submitting your manuscript to PLOS ONE. After careful consideration, we feel that it has merit but does not fully meet PLOS ONE’s publication criteria as it currently stands. Therefore, we invite you to submit a revised version of the manuscript that addresses the points raised during the review process.

Reviewers have commented on your paper. You will see that they are advising that you make some major revisions of your manuscript. If you are prepared to undertake the work required, I would be pleased to reconsider my decision.

We look forward to receiving your revised manuscript.

Kind regards,

Ta-Chung Chu

Academic Editor

PLOS ONE

Reviewers' comments:

Reviewer's Responses to Questions

**Comments to the Author**

1. Is the manuscript technically sound, and do the data support the conclusions?

Reviewer #1: Partly

Reviewer #2: Yes

2. Has the statistical analysis been performed appropriately and rigorously? 

Reviewer #1: Yes

Reviewer #2: Yes

3. Have the authors made all data underlying the findings in their manuscript fully available?

Reviewer #1: Yes

Reviewer #2: Yes

4. Is the manuscript presented in an intelligible fashion and written in standard English?

Reviewer #1: Yes

Reviewer #2: Yes

5. Review Comments to the Author

Reviewer #1: This paper used GA and multiple criteria decision-making method (TOPSIS) to investigate the design of CMS from CF and machine cell layout. I have some comments as follows:

1. The abstract should be rewritten. The weights of factors are 0.1, 0.3, 0.7, 0.5 and 0.9. The authors should introduce that how they were obtained.

2. Introduction should be written. The background, motivation, and innovation of the research question in this article should be introduced in detail.

3. The image of fitness value and iteration number should be provided to demonstrate the convergence of the algorithm.

4. The layout of some formulas should be corrected. For example, Eq. (7) and Eq. (8).

5. The number of digits retained after the decimal point should be uniform. For example, Table 7 and Table 8.

6. Insufficient research in the article and prospects for future work should be provided in Conclusion.

7. The layout of current version should be improved.

Reviewer #2: I am writing in response to the manuscript I was requested to review. Overall, the content is compelling and of good quality, however, upon careful examination, I have identified certain areas that require further clarification.

One of my primary concerns pertains to the use of the Technique for Order of Preference by Similarity to Ideal Solution (TOPSIS) over other Multiple Criteria Decision Making (MCDM) techniques. I would appreciate it if the author could elucidate on the rationale behind selecting TOPSIS, and why it was deemed more suitable for this study compared to other MCDM methodologies.

Moving on to Table 12, I noticed that part 6 is processed in 14, which then necessitates a move to a different cell. I would urge the author to propose a solution or an alternative approach to mitigate this issue of moving a part to another cell for another processing.

Upon reviewing the results, I found a striking resemblance to the outcomes presented in the C and R study [34]. In order to maintain the unique contribution of this paper, I recommend the author provide a comprehensive explanation detailing how their method diverges from the one in that study [34].

Every research method has its inherent limitations. To present a balanced view and enhance the credibility of the study, could the author shed some light on the limitations of their chosen method?

Finally, in the interest of demonstrating the potential applicability and the future scope of their method, I would suggest the author discuss the possible future directions. How could this method be further extended or improved upon? What are the potential advancements or applications that could stem from this research?

Thank you for considering my recommendations. I believe addressing these points will greatly enhance the depth and clarity of the manuscript, ultimately contributing to its overall robustness.

6. PLOS authors have the option to publish the peer review history of their article (what does this mean?). If published, this will include your full peer review and any attached files.

Reviewer #1: No

Reviewer #2: No

---

## [Author Response · Author response to Decision Letter 0]

26 Nov 2023

Reviewer #1:

1.The abstract should be rewritten. The weights of factors are 0.1, 0.3, 0.7, 0.5 and 0.9. The authors should introduce that how they were obtained.

Rep: Thank you for pointing this out. we explicitly introduce how these weights were obtained "A GA is employed to identify machine cells and part families based on Grouping Efficiency (GE) as a fitness function. In contrast to previous research, which considered grouping efficiency with a weight factor (q = 0.5), this study utilizes various weight factor values (0.1, 0.3, 0.7, 0.5, and 0.9)."

2. Introduction should be written. The background, motivation, and innovation of the research question in this article should be introduced in detail.

Rep: We acknowledge the suggestion to enhance the Introduction section. In the revised manuscript, we provided a more detailed presentation of the background, motivation, and innovation related to the research question addressed in this study.

3. The image of fitness value and iteration number should be provided to demonstrate the convergence of the algorithm.

Rep: Thank you for the valuable suggestion. In the revised manuscript, we included an image illustrating the convergence of the algorithm by depicting the fitness value against the generation number. "Fig 6. Fitness function curve at weight factor 0.3."

4. The layout of some formulas should be corrected. For example, Eq. (7) and Eq. (8).

Rep: We appreciate your attention to detail. We reviewed and corrected the layout of the formulas in the revised manuscript.

5. The number of digits retained after the decimal point should be uniform. For example, Table 7 and Table 8.

Rep: In the revised manuscript, we ensured uniformity in the number of digits retained after the decimal point.

6. Insufficient research in the article and prospects for future work should be provided in Conclusion.

Rep: We appreciate the feedback and acknowledge the need for additional research insights and prospects for future work in the Conclusion section. we incorporated a more comprehensive discussion to address this concern. “This approach can be applied to a wider range of CMS design issues, including but not limited to reliability, group scheduling, production planning, and alternative layouts (such as U-shaped, double-row, or others). Moreover, future research can focus on integrating environmental considerations into the design and optimization of cellular manufacturing systems. This can involve incorporating criteria related to energy consumption, waste reduction, and carbon footprint in the decision-making process, contributing to the development of environmentally conscious manufacturing practices.”

7. The layout of current version should be improved.

Rep: Thank you for your feedback. We carefully reviewed and implemented enhancements to ensure a better presentation. If you have any specific areas of concern or additional recommendations, please share them with us.

Reviewer #2:

1.One of my primary concerns pertains to the use of the Technique for Order of Preference by Similarity to Ideal Solution (TOPSIS) over other Multiple Criteria Decision Making (MCDM) techniques. I would appreciate it if the author could elucidate on the rationale behind selecting TOPSIS, and why it was deemed more suitable for this study compared to other MCDM methodologies.

Rep: Thank you for your insightful comments and concerns regarding the choice of the Technique for Order of Preference by Similarity to Ideal Solution (TOPSIS) in our study. We appreciate your attention to this aspect and would like to provide a comprehensive explanation for our selection.

“In order to evaluate the optimal solution using the Multi-Criteria Decision-Making (MCDM) method, many MCDM models can be used in optimal selection, such as TOPSIS, Preference Ranking Organization Method for Enrichment Evaluation (PROMETHEE), AHP, ELimination Et Choix Traduisant la REalite (ELECTRE), and so on [34]. The Similarity to an Ideal Solution (TOPSIS) technique is used in this paper because it is easier to use, doesn't have strict rules about how the data is distributed or the size of the sample, and is better for sorting sample data internally [35], this paper adopts the TOPSIS technique. Moreover, the selection of the TOPSIS technique was based on its previous successful use in resolving decision-making problems of a similar nature [36].”

2.Moving on to Table 12, I noticed that part 6 is processed in 14, which then necessitates a move to a different cell. I would urge the author to propose a solution or an alternative approach to mitigate this issue of moving a part to another cell for another processing.

Rep: Thank you for your careful observation and insightful comment on Table 12. “In Table 12 the parts (P2, P4, 66, and P8) are considered exceptional parts because of processed in more than one machine cell. exceptional parts can be eliminated by duplicating the bottleneck machine, redesigning parts, eliminating the bottleneck operations, and releasing the capacity of bottleneck machine.”

3.Upon reviewing the results, I found a striking resemblance to the outcomes presented in the C and R study [34]. In order to maintain the unique contribution of this paper, I recommend the author provide a comprehensive explanation detailing how their method diverges from the one in that study [34].

Rep: Our proposed approach differs from the methods mentioned in the comparison in its sensitivity to changing the weighting factor, as mentioned in the revised manuscript. “The findings indicate that the proposed approach outperforms or achieves results that are on par with those presented in the literature, which indicates the sensitivity of the proposed approach when taking a different value for the weighting factor to maximize grouping efficiency and grouping efficacy.”

4.Every research method has its inherent limitations. To present a balanced view and enhance the credibility of the study, could the author shed some light on the limitations of their chosen method?

Rep: Thank you for highlighting the importance of acknowledging the inherent limitations of our research approach. in this context we added the following text. “It's possible that the study's performance measurements are insufficient to fully capture the full scope of a manufacturing system's efficiency and productivity. Furthermore, reliability or machine breakdowns are not dealt with in this paper because all the machines are assumed to be in perfect condition. These are the limitations of this study.”

5.Finally, in the interest of demonstrating the potential applicability and the future scope of their method, I would suggest the author discuss the possible future directions. How could this method be further extended or improved upon? What are the potential advancements or applications that could stem from this research?

Rep: Thank you for your insightful suggestion regarding the discussion of the potential applicability and future scope of our approach. we included a dedicated section to discuss possible future directions for our approach as follow:

“Based on the results of this research, this approach can be applied to a wider range of CMS design issues, including but not limited to reliability, group scheduling, production planning, and alternative layouts (such as U-shaped, double-row, or others). Moreover, future research can focus on integrating environmental considerations into the design and optimization of cellular manufacturing systems. This can involve incorporating criteria related to energy consumption, waste reduction, and carbon footprint in the decision-making process, contributing to the development of environmentally conscious manufacturing practices.”

---

## [Decision Letter · Decision Letter 1]

7 Dec 2023

Cell Formation and Layout Design using Genetic Algorithm and TOPSIS: A Case Study of Hydraulic Industries State Company

PONE-D-23-26796R1

Dear Dr. Dhayef,

We’re pleased to inform you that your manuscript has been judged scientifically suitable for publication and will be formally accepted for publication once it meets all outstanding technical requirements.

Kind regards,

Ta-Chung Chu

Academic Editor

PLOS ONE

---

## [Editor Report · Acceptance letter]

19 Dec 2023

PONE-D-23-26796R1 

PLOS ONE

Dear Dr. Dhayef, 

I'm pleased to inform you that your manuscript has been deemed suitable for publication in PLOS ONE. Congratulations! Your manuscript is now being handed over to our production team.

Kind regards, 

on behalf of

Dr. Ta-Chung Chu 

Academic Editor

PLOS ONE